# A Citizen-Sensing-Based Digital Service for the Analysis of On-Site Post-Earthquake Messages

**Paolino Di Felice** [1,*] and **Michele Iessi** [2]

1    Department of Industrial and Information Engineering and Economics, University of L'Aquila,
     67100 L'Aquila, Italy
2    Department of Information Engineering, Computer Science and Mathematics, University of L'Aquila,
     67100 L'Aquila, Italy; iessimichele@gmail.com
*    Correspondence: paolino.difelice@univaq.it; Tel.: +39-320-423-2040

**Abstract:** The effectiveness of disaster response depends on the *correctness* and *timeliness* of data regarding the location and the impact of the event. These two issues are critical when the data come from citizens' tweets, since the automatic classification of disaster-related tweets suffers from many shortcomings. In this paper, we explore an approach based on *participatory sensing* (i.e., a subset of mobile crowdsourcing that emphasizes the active and intentional participation of citizens to collect data from the place where they live or work). We operate with the hypothesis of a "*friendly world*", that is by assuming that after a calamitous event, in the survivors prevails the feeling of helping those who suffer. The extraction, from the Twitter repository, of the few tweets relevant to the event of interest has a long processing time. With the aggravating circumstance in the phase that follows a severe earthquake, the elaboration of tweets clashes with the need to act promptly. Our proposal allows a huge reduction of the *processing time*. This goal is reached by introducing a *service* and a *mobile app*, the latter is an intermediate tool between Twitter and the citizens, suitable to assist them to write structured messages that act as surrogates of tweets. The article describes the *architecture of the software service* and the *steps* involved in the retrieval, from the Twitter server, of the messages coming from citizens living in the places hit by the earthquake; moreover, it details the storage of those messages into a geographical database and their processing using SQL.

**Keywords:** participatory sensing; Twitter; earthquake; asset; infrastructure; ranking

---

## 1. Introduction

Think of a territory where a severe seismic event has occurred. The consequences are well known: death and destruction of buildings and infrastructures. The phase that follows the seismic event is dramatically complex both for people residing in the hit territories and for disaster responders. These latter urgently need to have "an overall picture" of the affected areas in order to be able to organize the rescue operations. The most critical questions to be answered concern: what are the affected municipalities; which assets have been damaged (dwellings, public buildings, etc.), and which infrastructures (highways, bridges, etc.); and to what extent they have been damaged. The possibility of saving lives depends on the speed and effectiveness of first responders.

Among social media, Twitter is a valuable source of information because of its increasing popularity and simplicity of use, and because it allows gathering the voice of citizens "on site". Two case studies carried out by Middleton and Modafferi [1], showed that crisis maps generated from tweets can be compared to the post-event maps delivered by national civil protection authorities. The frame of the present study is the so-called *participatory sensing* [2], i.e., a subset of mobile crowdsourcing that

emphasizes the active and intentional participation of citizens to collect data from the place where they live or work (for such a reason also called *citizen sensing*).

The research described in this paper springs from two assumptions and perceives two goals; they are discussed in the following. We believe that disaster responders need *as little data as possible* that they can trust blindly, to set the *initial plan* for helping the population on-site. Immediately after an earthquake, a relevant task for disaster responders is the identification of messages coming from the hit territories. Unfortunately, "identifying the location from where the information is being disseminated is still a non-trivial issue", (Laylavi et al. [3], p. 2). de Bruijn et al. [4] have published an interesting paper about the geoparsing of groups of related tweets. Their method improves previous proposals devoted to the identification of the location of tweets; nevertheless, authors state that about 10% of the predicted locations are incorrect. Because of wrong data, disaster responders could take counterproductive actions that might exacerbate the negative impacts of the disaster.

We operate by assuming that after a calamitous event that has left behind death and destruction, in the survivors prevail the feeling of helping those who suffer and the desire not to increase confusion; in the following, we will refer to such an assumption as a *friendly world*. Coherently, in the paper, it is assumed that those who want to make a contribution to the rescue operations activate the geolocation option of their social media.

The total number of tweets sent per day is around 500 million. This is big data. Extracting from the Twitter repository the tweets pertinent to the event of interest has, necessarily, a heavy processing time (on this point, Shimauchi et al. [5], p. 93, said: " ... there are too many tweets circulating on the Internet which hinders timely extraction of useful information."). With the aggravating circumstance in the phase that follows a severe earthquake, the elaboration of tweets clashes with the need to act promptly. The reduction of the *processing time* was one of the objectives of our research.

The paper focuses on the *architecture of a software service* and its *development*, as well as a *mobile app* (the latter is an intermediate tool between Twitter and the citizens) suitable to assist them to write structured messages that act as surrogates of tweets. The paper describes, moreover, the *steps* involved in the retrieval, from the Twitter server, of the messages coming from citizens living in the places hit by the disaster and their storage into a geographical database. Those data are then processed by running SQL spatial queries.

The article is structured as follows. Section 2 is the related work. The purpose is to provide evidence of the growing emphasis placed on the dual role of citizens as users and, at the same time, as generators of valuable geographical data. It is a shared opinion that in order to make assessments adhering to the reality of the territory the direct involvement of citizens is necessary. Section 3 is structured into four subsections. Section 3.1 introduces the definitions and notations that will be used throughout the paper. Section 3.2 provides an overview of the *app* that the citizens should use for writing messages about either buildings or infrastructures damaged by the earthquake. Section 3.3 describes, at a high level of abstraction, the *Service* architecture. Section 3.4 discusses the steps that produce the many-fold outputs to be made available to disaster responders just after an earthquake has occurred. Section 4 presents an experimental case study; it is general, so it can be repeated elsewhere, anytime. Section 5 summarizes the paper.

## 2. Related Work

Twitter has played a relevant role in the hours and days immediately after catastrophic events since its advent and has been rapidly becoming the first point of contact for people wishing to seek or provide information about those situations (e.g., [5–8]). Luna and Pennock [9] conducted a review and analysis of the literature regarding the application of social media to emergency management. Twitter, in this context, can fill what South [10] described as "*the lack of immediate flow of information from people at the scene towards authorities or those who can provide help.*"

To use tweets to build the map of the affected areas and the map of the damages, it is essential to identify their provenance. The solution to this problem is immediate in cases where they are

"geotagged". Twitter allows the inclusion of the coordinates about the current location of the user (*geotagging*) as metadata inside the tweets. Despite today smartphones coming with GPS, only a small fraction of Twitter users turn on the location function of their smartphones, mainly to preserve privacy [11] and to prevent the battery from running out quickly [12]. There are many sources that confirm that up to now, a negligible percentage of tweets is geotagged. For example, in [1], we can read: "*The first event studied was hurricane Sandy (Oct 2012), which caused major flooding in New York and New Jersey. The tweet dataset covered 5 days, contained 597,022 tweets of which 4302 had a location mention (just the 0.7%).*" A study by Burton et al. [13] found that the prevalence of geolocated tweets was only about 3%; while Fuchs et al. [14] estimated that 1% of the tweets in Germany were geolocated.

Many studies have proposed solutions to the problem of determining the origin of tweets (e.g., [4,15,16]). Their approaches range from machine learning, statistics, probability, natural language processing to geographical information systems and surveying. Laylavi et al. [3] gave an overview of existing approaches. Unfortunately, "*identifying the location from where the information is being disseminated is still a non-trivial issue*", (Laylavi et al. [3], p. 2). Ribeiro and Pappa [17] compared 16 methods for user location inference. They found out that the inferences of most of them were in contrast while covering different sets of users. Generally speaking, the criticality is inherent in the fact that *all* the methods designed to try to find out the location of tweets process the text of the tweets themselves. On this point, Luna and Pennock ([9], p. 574) stated the following: "*Knowledge extraction from social media is challenging because of the unstructured nature of the data. Text including sarcasm, grammatical errors, special characters, and emoji's makes the analysis even more challenging*". Martínez-Rojas et al. ([18], p. 202) pointed out another criticality in the elaboration of the text of the tweets: "*for a given emergency, tweets can be found in several languages.*"

Laylavi et al., ([3], p. 4) stated the following about the results of processing the text of the tweets: "*in terms of average distance error, existing works achieve either the city-level granularity or the average distance error of over 200 km at best, which are too coarse and not sufficiently detailed for the emergency response domain.*" Laylavi et al. [3] have provided a significant improvement, compared with previous methods, to the problem of tweets' localization. In fact, they have shown that their method is able to infer the location of 87% of the tweets successfully at an average distance error of 12.2 km and a median distance error of 4.5 km.

The method proposed by Laylavi et al. represents an interesting result about the location of the origin of tweets, provided that the application for which it is intended to be used has no stringent precision constraints. An average distance error of 12.2 km, in fact, implies that the maximum error can reach values significantly bigger. This uncertainty might cause false positives and false negatives, and those errors would have an impact on the correctness of the map of the municipalities that have suffered damages. Italy, for example, is divided into about 8000 municipalities, of which 5500 have less than 5000 inhabitants. One thousand nine hundred forty municipalities have less than 1000 inhabitants and an average extension of just 42 km$^2$ (http://www.comuniverso.it/index.cfm?Comuni_inferiori_ai_1000_ab&menu=51). Di Felice [19] has calculated, for *all* Italian municipalities, that the average value of the maximum distance between the centroid of the polygon that shapes the boundary of the generic municipality and the boundary itself is just 5 km. It follows that for many municipalities, the centroid-boundary minimum distance decreases below this value. These data prove the statement above.

Another big issue that requires additional research is the *credibility* of the content of tweets. On this concern, Martínez-Rojas et al. ([18], p. 204) said the following: "*To take advantage of Twitter data as a source of information, the control of false information is increasingly important but still quite challenging. This is especially crucial in cases of emergencies where information in real time may be essential for the success of emergency management.*" On the same issue, Martínez-Rojas et al. ([18], p. 202) added the following: "*Regarding the propagation of false information, it is still a challenge the identification of malicious content spread during emergencies.*" Luna and Pennock [9] mentioned six different scenarios where outdated, inaccurate, or false information has been disseminated through social media applications, slowing

down the recovery efforts. As asserted, for instance, by Karlova and Fisher [20] "*if the information spread is outdated, inaccurate or false, participants could take counterproductive actions that might exacerbate the negative impacts of disasters.*" To overcome all the pitfalls mentioned above, de Albuquerque et al. [21] entrusted three different subjects with the delicate task of processing the content of tweets. They carried out personally the classification of tweets in three categories: "off topic", "on topic, not relevant", and "on topic, relevant". Obviously, this phase slows down the whole process of tweet processing.

Browsing the Net (see, for instance: https://www.omnicoreagency.com/twitter-statistics/), you can read the following Twitter statistics: total number of monthly active Twitter users: 326 million (last updated: 26 October 2018), number of Twitter daily active users: 100 million (last updated: 24 June 2018), percentage of Twitter users on mobile: 80% (last updated: 24 June 2018), total number of tweets sent per day: 500 million (last updated: 24 June 2018). This is big data. It follows that the extraction, from the Twitter repository, of the tweets pertinent to the event of interest has a processing time far from being negligible. With the aggravating circumstance in the phase that follows a severe earthquake, the elaboration of tweets clashes with the need to act promptly.

The idea of developing a *service* and an *app* has emerged to overcome *all* the shortcomings listed so far. The proposal takes a "shortcut" (described in the remainder of this section) for the extraction, from the messages written by citizens on-site, of *data we can trust* to be made available to the disaster responders. In this sense, we agree totally with the opinion of Budde et al. ([22], Footnote 3 on page 12): "*We disregard malicious users here, as we are convinced that someone determined to willingly submitting false data will find a way to do so.*" In order to avoid complex (and therefore slow) processing of the messages from the citizens, we assume that those who will use our *app* must know that their messages will be counted as reports of damaged assets or damaged infrastructures. In other words, we operate in the (already mentioned) hypothesis of a *friendly world*, whose natural frame is *participatory sensing* [2].

Our approach gives a central role to citizens that we trust, in a new and constructive vision that leads to the solution of real problems through a gainful collaboration among the various components of the society. Such a vision is in tune with the conclusions of the independent High Level Group on maximizing the impact of EU Research & Innovation Programmes, which recommends a greater mobilization and involvement of citizens in the future (the Lab-Fab-App report, [23]). The need of collaboration with citizens in solving real problems is strengthened in *citizen science* where the cooperation invests research [24].

The empirical results of the study carried out by Wamuyu [25] showed that there is an interest among low-income urban communities in social media-based communication for civic online participation. He concluded the paper as follows: "*This is a fertile ground for those seeking to drive the e-participation agenda both in politics and in civil responsibilities and to enhance community involvement among the low-income urban communities.*" Policy makers and (national and local) emergency agencies should take a clue from studies such as [25] to come-up with emergency policies that put greater responsibility and participation on people. The assumption of a *friendly world*, at the base of our study, embeds this suggestion.

In the Abstract of [2], Khoi and Casteleyn wrote the following: "*Participatory sensing applications are now able to effectively collect a variety of information types with high accuracy. Success, nevertheless, depends largely on the active participation of the users.*" Obviously, we agree with this statement, but our opinion is that besides voluntary participation, another element is necessary to reach good results from the elaboration of the messages: a *correct* and *aware* participation by citizens. The assumption of a *friendly world* comes from that.

Many researches have highlighted which factors determine the positive engagement of people in highly dramatic situations. Bekkers and Wiepking [26] identified eight mechanisms that drive charitable giving: (a) awareness of need; (b) solicitation; (c) costs and benefits; (d) altruism; (e) reputation; (f) psychological benefits; (g) values; (h) efficacy. Three of them can be translated to the context of natural disasters, namely: *awareness of need*, *solicitation*, and *altruism*. Sentences that support this statement close this section.

*Awareness of need:* "The severity of needs increases volunteering." "Making people aware of the need for contributions increases volunteering." The hours immediately after a natural disaster are characterized by an enormous need of help from all people of good will. Kalish [27] investigated the relationship between natural disasters and volunteerism. The findings of such a study highlighted the importance of policy focused on harnessing volunteer labor in the wake of natural disasters.

*Solicitation:* Asking people directly increases volunteering. [26]. *"The mere fact of being asked to volunteer greatly increases the likelihood that people start to volunteer"*, (Bekkers and de Wit [28]). We believe that people being solicited/educated that take part actively in the post phase of an earthquake in the case of an actual occurrence would react positively.

*Altruism:* *"The hypothesis on altruism is that when the material, social or psychological needs of recipients are more severe, people will volunteer more."*, Bekkers and de Wit ([28], p. 11). Tierney, Lindell, and Perry [29] asserted that "*altruism is evident in immediate volunteer work after natural disasters and in emergencies."*

## 3. Materials and Methods

### 3.1. Definitions and Notations

This section introduces the main definitions and notations used in the paper.

1. $\mathcal{T}$ (*Territory of interest*) denotes a geographic area, while $\mathcal{T}_i$ ($i = 1, 2, ...,$) denotes its internal administrative subdivisions. Italy, for instance, is structured at three different levels: municipalities, provinces, and regions (moving from the bottom to the top).
2. *Target* is either an asset or an infrastructure located inside $\mathcal{T}$. *Asset* and *Infrastructure* denote, respectively, the set of assets and the set of infrastructures in $\mathcal{T}$. Relevant assets are dwellings and public buildings (as, for instance, schools, hospitals, banks, government offices, post office, police stations, malls, etc.). Relevant infrastructures include roads, highways, bridges, and railways. Each target is defined by the tuple $\langle ID, description, geometry\rangle$, *ID* being an identifying code.
3. *GeoReport* is a warning about a serious damage caused by the earthquake to a specific *Target*, issued by a user of the *Service* we are going to present. To each *GeoReport* is associated a latitude and longitude pair of coordinates. *R* denotes the set of all *GeoReports* with the same hashtag (#*defaultHashtag*). In the following, the string *defaultHashtag* is called the *default hashtag*.
4. *The score of subdivision* $\mathcal{T}_i$ is the integer that denotes the number of *reports* issued from within $\mathcal{T}_i$; while the *score of a generic target* in *Target* is the integer denoting the number of *GeoReports* regarding the target.
5. *Thesaurus* is a rich collection of synonyms and common abbreviations about the targets in $\mathcal{T}$.

### 3.2. The App Template

This article proposes a *Service* and a mobile *app*. The *Service* will be discussed in the next two sections; below, we focus on the basic elements of the *app* through which citizens can write *GeoReports*. The template of Figure 1 brings together the information that can be part of a generic *GeoReport*. As you can see, the *app* is structured in terms of a certain number (of predefined) keywords.

The #defaultHashtag string has to be shared by *all* the citizens living in the territories hit by the earthquake in order to simplify and speed up the retrieval (from the Twitter repository) of the *GeoReports* coming from those areas. Section 3.3 details *who* set such a string and *how*. Isolating *all* and *only* the tweets relevant to a specific natural disaster is a very complex operation due to the huge number of tweets collected in the Twitter repository, of which only a small part come from the region affected by the event. The high number of different hashtags used in tweets coming from the impact region is a further critical issue to be taken into account. Murzintcev and Cheng [30], for example, extracted 4314 hashtags from the filtered tweets coming from two USA regions affected by two disastrous floods in 2013. To try to cope with those issues, Murzintcev and Cheng proposed a hashtag-based data retrieval method that uses, in sequence, up to four different filters. Nevertheless,

their information retrieval method fails in several ways. Just an example, since not every tweet contains hashtags, therefore, when using their method, those messages are lost.

The field `YOUR COORDINATES` contains the geographic coordinates of the position of the person who sent the *GeoReport*.

| `#defaultHashtag` | | |
|---|---|---|
| `YOUR COORDINATES` | | |
| `Damaged Asset` | Bank | `<add image>` |
| | Church | |
| | Dwelling | |
| | Hospital | |
| | Mall | |
| | Public Office | |
| | School | |
| | University | |
| `Damaged Infrastructure` | Bridge | `<add image>` |
| | Highway | |
| | Local road | |
| | Railway | |
| `Comment` | | |

**Figure 1.** The *app* template.

Each *GeoReport* can concern only one warning, written by selecting either the field `Damaged Asset` or `Damaged Infrastructure`. If both those fields are empty, then the *GeoReport* will be ignored (see, Section 3.3). The `Comment` field has to be used to enter the name and address of the damaged asset or the damaged infrastructure name. It follows that the processing of the citizens' reports (charged to the *Service*) is cross-language.

The automatic classification of disaster-related tweets is a relevant and complex problem, still open, addressed by many researchers (e.g., [31–36]). About the difficulties to be overcome, Stowe, K. et al. ([33], p. 1) wrote: "*Identifying relevant information in social media is challenging due to the low signal-to-noise ratio.*" To et al. [34], on the same point, said: "*Even though abundant tweets are promising as a data source, it is challenging to automatically identify relevant messages since tweet are short and unstructured, resulting to unsatisfactory classification performance of conventional learning-based approaches.*" Moreover, Asakura, Hangyo, and Komachi ([36], p. 24) clarified, through examples, that: "*Collecting posts including keywords about a disaster is not sufficient to judge whether the disaster actually occurred.*" The predefined format of our *app* (Figure 1) implies that the processing time charged to the underlying *Service* is low, while the quality of the result is high. In particular, *precision* and *recall* are both equal to 1. This result comes from the assumption of a *friendly world* and from the adoption of the shared hashtag (`#defaultHashtag`) by the citizens participating in the post-earthquake support action. These two conditions imply the retrieval of *all* and *only* the *GeoReports* written by the citizens participating in the sensing. In numbers, it follows that: P(recision) = TP/(TP + FP) = TP/(TP + 0) and R(ecall) = TP/(TP + FN) = TP/(TP+ 0) where TP = True Positives, FP = False Positives, and FN = False Negatives.

### 3.3. Architecture of the Software Service

Figure 2 sketches the architecture of the implemented *Service*. In the past, other authors have pursued a similar goal (e.g., [37–39]). Shatabda [38], for example, proposed a framework based, as in our case, on participatory sensing. The aim of his software system was the monitoring of pollution in urban areas of Bangladesh. The paper by Oussalah et al. [37], on the contrary, described the design and implementation of a software system suitable to collect geolocated Twitter data live and carry out various search functionalities. The differences of our work with theirs are many, and all stem from the adoption of the *friendly world* hypothesis. Both papers assume that the messages are geolocated.

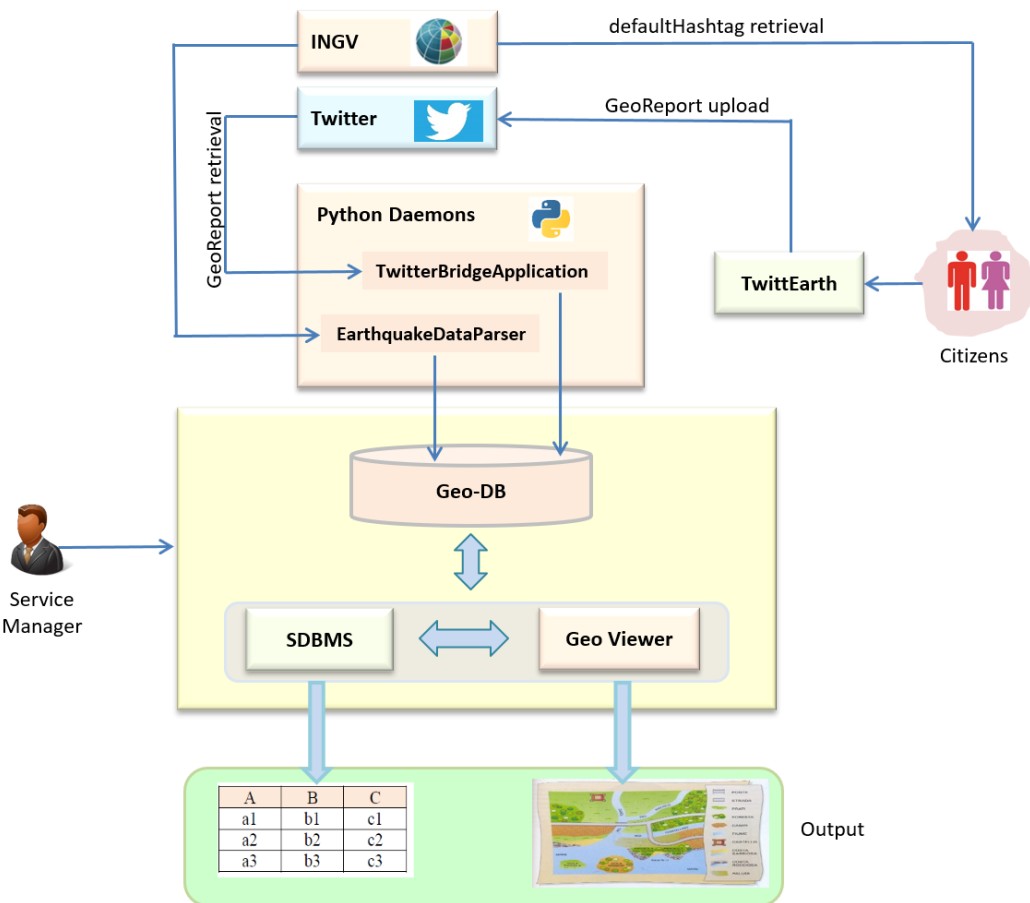

**Figure 2.** Architecture of the implemented *Service*. SDBMS stands for Spatial Database Management System.

The proposed *Service* involves two stakeholders: the *Service Manager* and the citizens. The former is responsible for the activation of the *Service* and its use. The successful activation of the *Service* and its effective use require that the Manager holds an IT master program degree with specific skills in *database administration*. The role of Service Manager can be covered either by a referent of a government agency (for example, the Department of Civil Protection) or by a voluntary association. According to the Union of International Associations [40], there were over 38,000 international nonprofits in 2014, providing disaster relief and delivering social services. The activation of the *Service* is a complex and quite long operation, to be carried out only once and a long time before the earthquake may occur. It consists of creating a multi-table database, as will be detailed in the following.

The architecture of the *Service* of Figure 2 is made up of the following software components: (a) *EarthquakeDataParser*; (b) *TwittEarth*; (c) *TwitterBridgeApplication*; and (d) *Spatial Database Management System*. Below, we describe all of them in sequence.

### 3.3.1. EarthquakeDataParser

The *EarthquakeDataParser* is a Python daemon, which connects to an authoritative data source (the National Institute of Geophysics and Volcanology in the case of Italy: INGV, Istituto Nazionale di Geofisica e Vulcanologia) web service and downloads (in the JSON format) the magnitude, epicenter, and timestamp of the earthquake matching the input parameters (namely, the name of the subdivision where the earthquake occurred and the approximate value of its magnitude). The returned data are uploaded onto the Geographical DataBase (Geo-DB). The data about the earthquake enrich the semantics of the maps that will be built using the proposed *Service*.

3.3.2. TwittEarth

*TwittEarth* is an Android application that allows citizens to publish *GeoReports* having the predefined format of Section 3.2. *TwittEarth* performs the "landing" of the *GeoReports* written by the citizens in the Twitter repository using the Twitter Kit Java library for Android. Twitter Kit acts as a proxy for using Twitter's REST APIs. In summary, *TwittEarth* is an intermediate tool between Twitter and the citizens, suitable to assist them to write structured messages that act as surrogates of tweets. The reasons for relying on Twitter, instead of setting up a standalone infrastructure, are explained below:

- The reliability of the communications network is vital especially during severe events. Often, during large events, the network fails because of high traffic. Of course, Twitter is not an exception, but its downtime tends to zero because its infrastructure and software are very robust;
- Twitter has a public set of robust APIs that developers can reach by means of HTTP requests. These APIs allow the retrieval and manipulation of tweets stored in the social network repository, as well as sending new messages. Moreover, the APIs support the formulation of advanced queries based on text, hashtag, location, or specific people. These services promise high performance. Twitter requires that any Twitter app authenticates first.

Figure 3 shows the initial screen of the *TwittEarth* app. The *app* permissions are:

- *Read and Write grant*: in order to be able to post *GeoReports*;
- *Location*: needed to access the last known position of the device (GPS, WiFi, or cellular) in order to geolocalize the *GeoReport*;
- *Write to External Storage*: needed to store the pictures.

Citizens need a Twitter account to be able to send messages through *TwittEarth*. After logging into the social network (Figure 3), the user may post the text of the report (at most 280 characters).

The first two images of Figure 4 depict (in sequence) the screen of *TwittEarth* to be used to report a damage concerning an asset or an infrastructure; while the third image shows the photo of a damaged building.

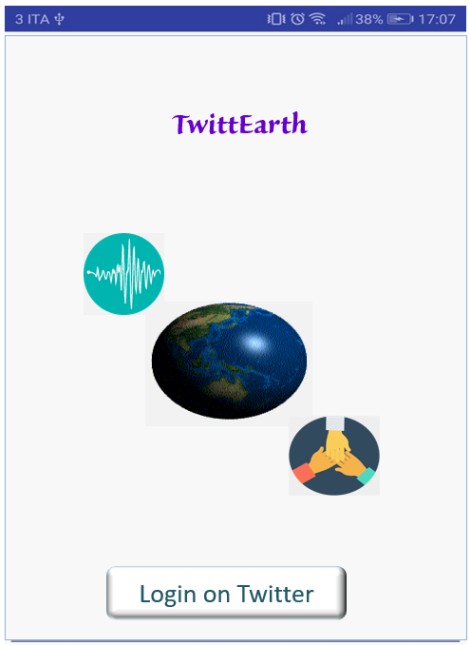

**Figure 3.** The initial screen of the TwittEarth mobile app.

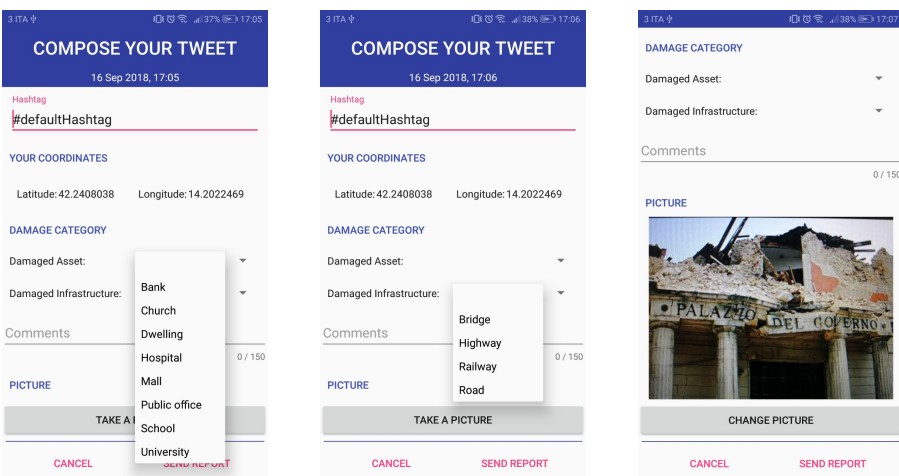

**Figure 4.** A close look at the TwittEarth app.

### 3.3.3. TwitterBridgeApplication

The *TwitterBridgeApplication* is a Python daemon. At the start time, it connects to Twitter through the Twitter Search APIs. Implementing the *Service*, it was necessary to make recourse to the *Premium search APIs* (https://developer.twitter.com/en/docs/tweets/search/api-reference), because not all the tweets are made available via the Standard search APIs. Specifically, we referred to the *Search Tweets: 30-day endpoint*. This option allows accessing tweets posted within the last 30 days. The restriction to this time window does not represent a limitation for our application since its maximum utility is on the hours and days that follow the earthquake; moreover, it provides the benefit of drastically reducing the number of tweets to look into, with an obvious speed up of the overall processing time.

Once the set time is matched (every 15 min, configurable), the *GeoReports* having the input hashtag `#defaultHashtag` are fetched in the JSON format. As the second step, the daemon extracts the location (i.e., the latitude and longitude) from the *GeoReports* that are geolocalized; finally, the filtered messages are uploaded onto the Geo-DB. This loop continues until it is manually shut down. Figure 5 sketches the three steps.

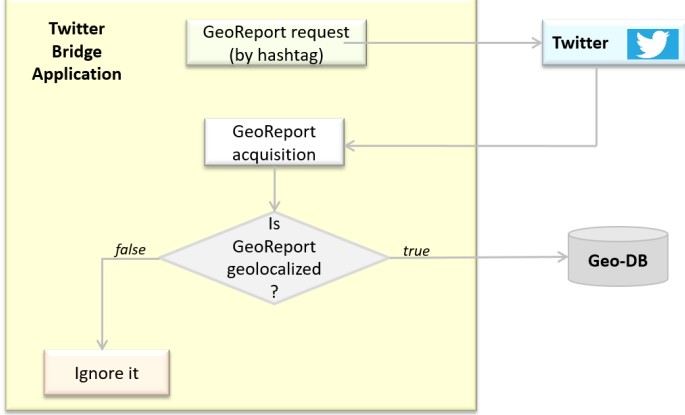

**Figure 5.** Functional architecture of the TwitterBridgeApplication.

As mentioned, the architecture of Figure 2 attributes the management of the proposed *Service* to the INGV. In such a hypothesis, the Service Manager has to add to the INGV website that lists recent earthquakes (Figure 6) a further column about the value of the string that instantiates the keyword `defaultHashtag`. The choice of this word must allow the unambiguous identification (by *TwitterBridgeApplication*) of the *GeoReports* about a specific seismic event. For example, the string could

be composed of the name of the place affected by the earthquake followed by the timestamp about the time of the adverse event. Citizens on-site, before sending their messages about observed damages, will have to read this string by accessing the INGV site from their mobile device.

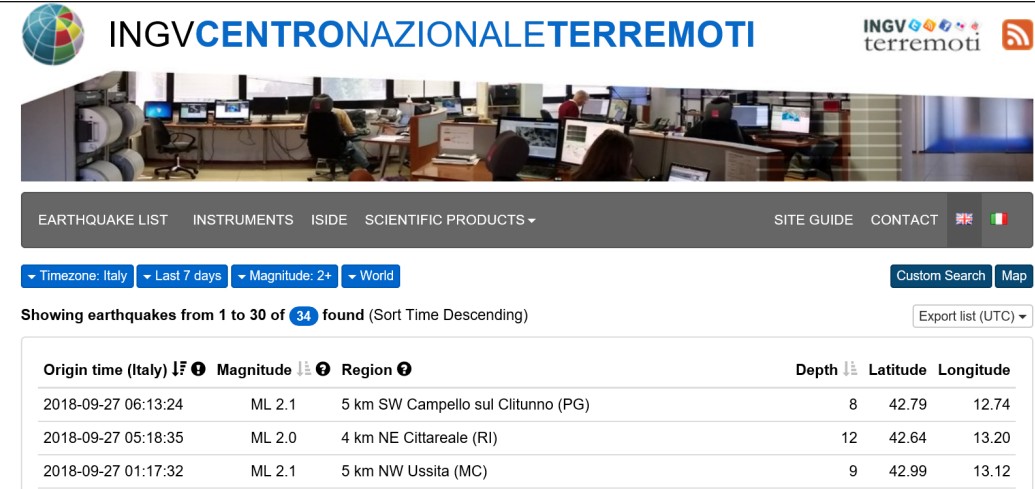

**Figure 6.** The INGV page about recent earthquakes (http://cnt.rm.ingv.it/).

### 3.3.4. Spatial Database Management System

The *Spatial Database Management System* is the core of the architecture because it manages the storage of *all* the input data and also because *all* the computations on the content of *GeoReports* finalized to the production of the many-fold outputs are implemented as SQL views that call powerful PL/pgSQL User-Defined Functions (UDFs).

In detail, UDFs have been written for:

- carrying out the census of all administrative subdivisions that suffered severe damages from the earthquake, thus delimiting the boundary of the damage area;
- building interactive maps that show the targets for which a severe damage has been reported;
- building the ranking of the various administrative units with respect to the damages they suffered;
- computing the ranking of which categories of assets (e.g., public buildings or churches) and which infrastructures (e.g., highways instead of roads) have suffered the greatest damages;
- returning tables about the ranking of targets for which severe damages have been reported and others that specify the geographical position (in terms of latitude and longitude) of these targets.

In summary, as shown in Figure 2, the proposed *Service* returns results both in the tabular metaphor (typical of relational databases) and in the map metaphor (characteristic of the GIS world).

Table 1 shows the mapping of the theoretical concepts into database entities.

**Table 1.** Entity mapping.

| Definition | Entity |
|---|---|
| *GeoReports* | GeoReports |
| $\mathcal{T}$ | Subdivisions |
| *Assets* | Assets |
| *Infrastructures* | Infrastructures |
| *Thesaurus* | Thesaurus |
| *Earthquake* | Earthquakes |

Figure 7 shows the tables of the correspondent PostgreSQL Geo-DB. In the image, the check marks highlight the primary key of the tables, while each arc connecting two tables denotes a *join path* between them.

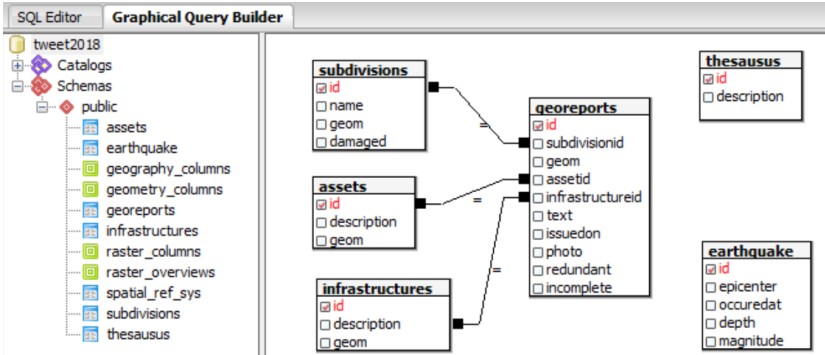

**Figure 7.** The Geo-DB tables and their relationships.

Assets and infrastructures are kept in two distinct tables because, very often, they are huge (order of gigabytes), collect input data from heterogeneous data sources, and are described by different features. Moreover, it is a common practice to model assets as `points` and infrastructures as `lines`. The geometries of geographical features are expressed in the WGS84 reference system (PostGIS code: 4326). The attribute `description` of tables `assets`, `infrastructures`, and `thesaurus` is of type `json`; attribute `text` of table `georeports` is of type JSON, as well. The attribute `photo` of table `georeports` is of type `raster`, so it is suitable to store the image in the citizens' messages (if any).

The `thesaurus` serves to identify the damaged target mentioned in the `Comment` field of the report. This need arises because each point of interest (e.g., mall, restaurant, theater, school, etc.) has a name and is described by an address, but in real life, it often happens that the name and address of a point of interest are shortened by citizens living on-site, and the address, sometimes, is even dropped. The thesaurus about the targets of a territory must contain within it, for each element, the name and full address, besides the richest possible list of "alternative abbreviations" used in everyday routine. The thesaurus collects the alphanumeric data about the targets that are in the reference territory. The geometry of those targets, vice versa, is stored in the tables `assets` and `infrastructures` whose (descriptive and geometric) content can be downloaded from accredited institutions.

Let us refer, for example, to the city of L'Aquila. It is the region capital of Abruzzo (Central Italy). In L'Aquila, there is an important hospital called San Salvatore, located in Columbus Square. In the daily life of the city residents, it is frequent to refer to this public structure with one of the following abbreviations: San Salvatore hospital of L'Aquila; San Salvatore hospital; hospital of L'Aquila; the regional hospital; and San Salvatore. It is infrequent that residents, talking about the hospital, mention its full address. Figure 8 shows the structure and content of the JSON record (inside the `thesaurus` table) collecting the name, address, and abbreviations of the L'Aquila hospital. (The layout of the record in the figure facilitates the understanding of the JSON data, generally complex, but PostgreSQL requires that the text be written on a single line.)

```
INSERT INTO thesaurus (description) VALUES (
'{
  "assetName":
    {"Name":      "San Salvatore hospital",
     "nickname1": "Hospital of L'Aquila",
     "nickname2": "The San Salvatore",
     "nickname3": "The regional hospital"},
  "assetAddress":
    {"Address":   "Columbus Square, 67100 L'Aquila",
     "nickname1": "Columbus Square",
     "nickname2": "Columbus Square, L'Aquila"}
}' );
```

**Figure 8.** The (JSON) record about the hospital of L'Aquila.

The construction of an effective thesaurus is a long and expensive operation, and building a "complete" one is an ambitious goal. The only way that seems to be feasible to contain the time and the cost involved in the construction of the thesaurus of the administrative subdivisions of the various states is to involve the residents.

### 3.4. The Computation Steps

Figure 9 lists the (eight) steps that produce the many-fold outputs that can be made available to disaster responders just after an earthquake has occurred. This section focuses on each of them.

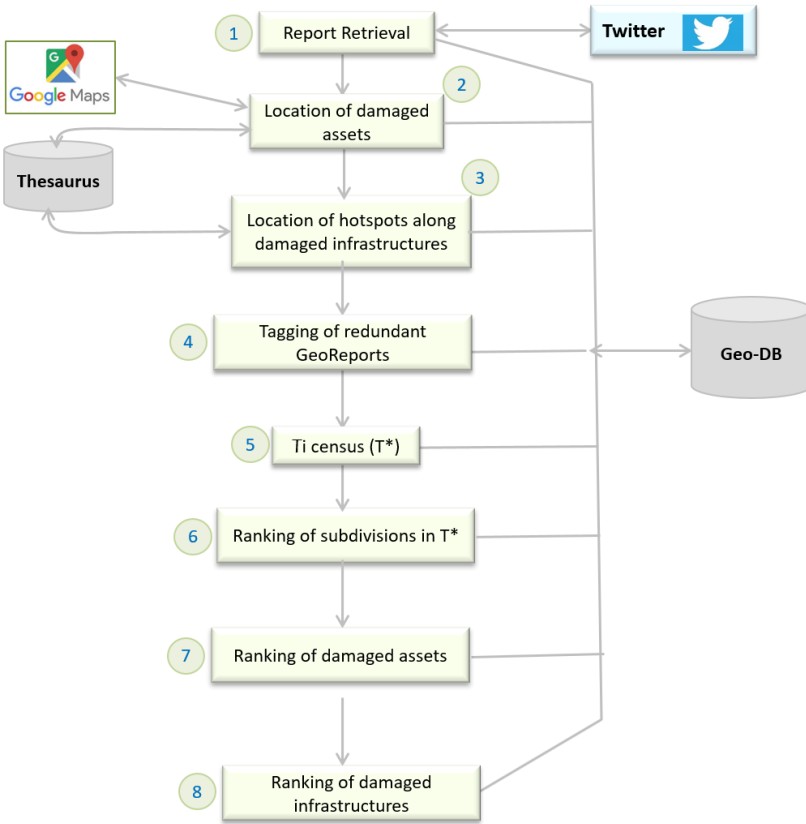

**Figure 9.** The steps to be computed.

The **first step** is devoted to the retrieval of citizens' reports from the Twitter repository. Messages that do not contain the reference hashtag (#defaultHashtag) in their body are skipped. Then, the software checks whether those messages are geotagged. A PL/pgSQL UDF (getLocation()) extracts the latitude and longitude from the citizen's reports and stores them into the attribute geom of table georeports. The knowledge of the coordinates allows the computation of the administrative subdivisions containing those locations; hence, the subdivisionID field is updated. The remaining data (in the selected messages) are stored in the attribute text of table georeports, in the JSON format (Figure 10). The JSON keys Target, Category, and Comment correspond to the *GeoReport*'s metadata in Figure 10; while Figure 11 shows the values of those keys extracted from a message reporting that the hospital of L'Aquila has been damaged.

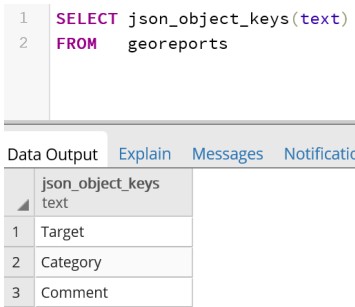

**Figure 10.** The JSON keys composing the attribute `text` of table `georeports`.

```
1  INSERT INTO georeports(text) VALUES (
2  '{
3    "Target":    "asset",
4    "Category":  "hospital",
5    "Comment":   "San Salvatore hospital, Columbus Square, L'Aquila"
6  }' );
```

**Figure 11.** An instance of the three JSON keys.

The **second** and the **third step** concern the computation of the location of assets and infrastructures severely damaged by the earthquake. The geographical location of these targets is essential for the disaster responders to organize the first on-site rescue actions. The difference that exists in the geometry of assets with respect to infrastructures (the former are modeled as points, while the latter as lines) requires that these two types of targets are treated separately.

**Location of damaged assets**

The initial goal of the *Service* is to complete (if necessary) and standardize the *name* and *address* of the asset mentioned in the JSON key `Comment` stored in the attribute `text` of the `georeports` table. The format of the output string is the following: $\langle [assetName];$ $[streetName]|[squareName]; number; postalCode; town \rangle$. By taking into account the values of the JSON keys `Target` and `Category` (`asset` and `hospital` in the example of Figure 11), the *Service* can limit the comparison of the string into the JSON key `Comment` against the small subset of the elements in the thesaurus that match the values of those two keys.

The main steps of the UDF that builds the string about the *name* and *address* of the asset whose geographical position is to be identified are listed below:

- extraction of the string linked to the JSON key `Comment` of column `text` of table `georeports`. The PostgreSQL query is: `SELECT to_jsonb(text -» 'Comment') AS assetByCitizen FROM georeports`. With regard to the warning of Figure 11, such a query returns the following (JSONB) value: "San Salvatore hospital, Columbus Square, L'Aquila";
- access to table `thesaurus`; (a) extraction of the values linked to the JSON keys `AssetName` and `AssetAddress` (PostgreSQL operator: '->'); (b) transformation of those strings in the JSONB format (PostgreSQL function: 'to_jsonb()'); (c) concatenation of those two strings (PostgreSQL operator: '||'; available only on JSONB data). Steps (a), (b), and (c) are repeated for all the "name-address" alternative descriptions listed in the thesaurus. With regard to the example of Figure 8, the alternatives are 4 × 3; Figure 12 shows the SQL query that returns one of those strings;
- computation of the "similarity" of the values of the strings `assetByCitizen` and `assetFromThesaurus` (i.e., "San Salvatore hospital, Columbus Square, L'Aquila" vs. "San Salvatore hospital", "Columbus Square, 67100 L'Aquila") by means of equation: $S = 2 \times m/n$, proposed by Chen and Lim [41] to measure the similarity of pairs of tweets ($n$ is the total number of words in the two tweets, while $m$ is the number of identical words of tweets). In [41], two tweets are recognized as nearly identical when $S$ exceeds the threshold of 0.9 (in the case of the example, $S = 2 \times 6/13 = 0.92$). Our UDF adopts the same threshold.

```
1  SELECT
2      (to_jsonb(description -> 'assetName'    -> 'Name')    ||
3       to_jsonb(description -> 'assetAddress' -> 'Address'))
4              AS  assetFromThesaurus
5  FROM   thesaurus;
```

Data Output | Explain | Messages | Notifications | Query History

| assetfromthesaurus |
| jsonb |
|---|
| 1 | [San Salvatore hospital,Columbus Square, 67100 L'Aquila] |

**Figure 12.** The (JSONB) value about the hospital of L'Aquila.

The *Service* calculates the geographical position of an asset, mentioned in a *GeoReport*, following two distinct paths and then comparing the result (Figure 13).

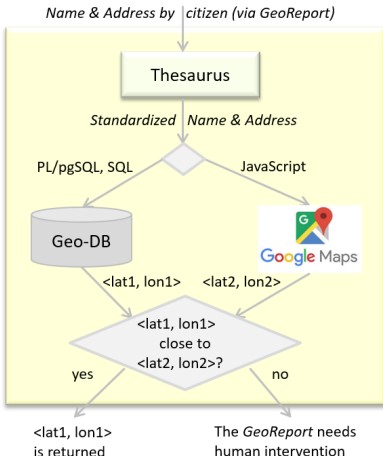

**Figure 13.** Geographical location of damaged assets.

**Case 1.** The (enhanced and standardized) name and address of the asset are *complete*, i.e., they do not have a `NULL` value for the compatible fields listed above. First (left path of Figure 13), such a string is compared with the values of column `description` of table `assets` of the Geo-DB (implementation: a SQL view that calls a PL/pgSQL UDF). Once the tuple has been identified, the content of the column `geom` (that is, a pair of coordinates in the WGS84 SRS) is extracted. Let us denote the output as $\langle \texttt{lat1,lon1} \rangle$.

Then (right path of Figure 13), the address of the asset is used to start the *geocoding* operation ([42] describes best practices about geocoding) with the help of the API of Google Maps (implementation: a JavaScript that calls function `geocodeAddress()`). Let us denote the output as $\langle \texttt{lat2,lon2} \rangle$.

If the distance between the points $\langle \texttt{lat1,lon1} \rangle$ and $\langle \texttt{lat2,lon2} \rangle$ is a few meters, then it is considered that the identification of the target is successful, and the *Service* returns $\langle \texttt{lat1,lon1} \rangle$ as the position of the damaged asset. Otherwise, the processed message is put on a list of warnings for which the *Service* reports that a human intervention is required (on the charge of the Service Manager and/or by some member of the disaster responder team).

**Case 2.** The (enhanced and standardized) name and address of the asset are *not complete*. The causes of incompleteness are multiple. Let us suppose, for example, that the citizen observes a partially collapsed school building. Since he/she is from the place, it is likely to assume that he/she knows that it is a school. It is not reasonable, however, to assume that he/she knows the name and the address of the school or that, even knowing them, he/she necessarily remembers them in a situation of high emotional stress like that which takes possession of people immediately after having experienced a devastating earthquake. It is, therefore, reasonable to think that he/she chooses, without uncertainty, *school* among the options listed by the TwittEarth app, but then, he/she does not write anything else in the `Comment` field of the message or he/she writes, for example, *elementary school of L'Aquila*, but

does not provide the exact name (say Dante Alighieri elementary school), nor the full address (Robert Kennedy Road 45, 67100 L'Aquila).

The *GeoReports* that fall into this category are marked "incomplete" (attribute `incomplete` in table `georeports` of the Geo-DB is set to `true`).

**Location of hotspots along damaged infrastructures (Step 3)**

The *Service* performs the actions described below for each citizen's report:

- It standardizes (by accessing the thesaurus) the name of the target mentioned in the message;
- It retrieves the tuple about the target at hand from table `infrastructure`. The geometry (of type `line` and stored in the `geom` field) characterizes an infrastructure. The length of a line can range from a few kilometers to hundreds of kilometers (this is the case, for example, of the highways). In order to facilitate the work of disaster responders, it is necessary to provide them with information about the stretch of the line (briefly *hotspot*) that has been damaged. The algorithm for calculating the hotspots along an infrastructure consists of the following steps:

  - Trace the straight line passing through the point that denotes the position of the citizen (who sent the message) and that is orthogonal to the geometry of the infrastructure at hand;
  - Calculate the geographical coordinates of the point (say *H*) in which the straight line intersects the infrastructure. This point is at the minimum distance from the person who wrote the message;
  - Return a buffer centered on point *H* and lying along the geometry of the infrastructure.

The **fourth step** (of Figure 9) tags as *redundant* the *GeoReports* about either the same asset or the same hotspot of an infrastructure. These two cases are discussed in sequence.

Notice that redundant *GeoReports* are not dropped from the Geo-DB because they carry double information. Suppose, for example, that the same target is mentioned in *N* messages. This confirms that the target is effectively damaged. Moreover, the greater the value of *N*, the greater the severity of the damage; the severity that could also not be disjoined from the relevance of the involved target for the territory where it is (for example, the collapse of a bridge).

**Case 1.** The developed *Service* identifies the *GeoReports* coming from the same $\mathcal{T}_i$ and concerning the same asset. In order to make a correct census of the damage suffered by the various administrative units and to draw up the correct ranking of the affected assets, it is necessary to isolate these repetitions.

As the first option, the possibility of using the method adopted by Chen and Lim [41] to measure the *similarity* of pairs of *GeoReports* was evaluated. In our case, this approach does not work all the time, as explained below. Let us refer to the following two messages:

*GeoReport* 1: Asset; Dwelling; Road Alessandro Magno, 10, 00130 Rome
*GeoReport* 2: Asset; Dwelling; Road Alessandro Magno, 21, 00130 Rome
$S = 0.875$. Therefore, the two warnings are correctly kept.

Now, let us refer to two more *GeoReports*:

*GeoReport* 3: Asset; Dwelling; Road Alessandro Magno, 00130 Rome
*GeoReport* 4: Asset; Dwelling; Road Alessandro Magno, 00130 Rome
$S = 1$. Therefore, *GeoReport* 4 is considered a duplicate, and accordingly, it has to be tagged as redundant.

Unfortunately, this may turn into an incautious loss of information in case the two warnings actually refer to two different dwellings located along the same road, but the two citizens who sent the reports simply were unable to read the house number. Below, we present our strategy.

If, in the two *GeoReports*, the asset name and address coincide ($S = 1$), then the report written per second is marked *redundant* (by writing `true` in the field `redundant` of table `georeports`); otherwise,

that message is marked *redundant* only if the Euclidean distance between the position of the citizens who wrote the two reports is a matter of a few meters.

**Case 2.** A *GeoReport* is marked as redundant only if the following two conditions are both true: (a) it reports about a hotspot belonging to an infrastructure already mentioned in a previous report; (b) the Euclidean distance between the two hotspots is a few meters (Step 3).

The **fifth step** identifies the administrative subdivisions in $\mathcal{T}$ from which at least one geotagged report has been written by citizens. $\mathcal{T}^*$ denotes such a subset of $\mathcal{T}$. The computation of $\mathcal{T}^*$ takes into account *incomplete GeoReports* as well (Case 2, Step 3). This step is implemented as a SQL spatial query (that calls an ad hoc PL/pgSQL UDF), which updates to `true` the initial value (`false`) of the column `damaged` of table `subdivisions` of the Geo-DB. The output of this step is the *first result* that can be achieved by using the proposed *Service*.

**Step 6** (of Figure 9) computes the ranking of the damaged administrative subdivisions. The main task of this step concerns the computation of the score of subdivisions in $\mathcal{T}^*$ (Section 3.1); that is, the counting of how many non-redundant reports have been written by citizens about the damages that they had the chance to encounter. The ranking of the administrative subdivisions $\mathcal{T}_i$ in $\mathcal{T}^*$ is obtained by sorting, in decreasing order, the score values. This ranking is the *second result* that can be returned to the disaster responders.

**Step 7** and **Step 8** compute, in order, the ranking about the assets and the infrastructures in $\mathcal{T}^*$, by counting the number of non-redundant reports present in the Geo-DB when the processing is started (the *third* and the fourth *result*). This calculation translates into a spatial SQL query that benefits from the "join path" that exists between the tables `georeports`, `assets`, and `infrastructures`.

## 4. A Case Study

We have conducted a few preliminary experiments with the proposed *Service* by referring to the Abruzzo region (center of Italy), a territory repeatedly and severely affected by destructive earthquakes. Consider that the seismic event of 2009 killed 309 people, injured over 1600, made 60,000 homeless, and caused over 10 billion euros of estimated damage. Abruzzo consists of four provinces (L'Aquila, Teramo, Pescara, and Chieti) and 308 municipalities (Figure 14). The inhabitants are 1,315,196. The experiments consisted of supposing that a destructive earthquake had occurred inside the municipality of L'Aquila.

The case study offers the chance to show the kind of results (i.e., maps and tables) that the Service Manager can make available to the disaster responders and local authorities, in the hours that immediately follow the seismic event. The aim is to "touch" the practical utility of those decision support tools. The emulation of the behavior of the citizens after an earthquake is not in the focus of our research, and hence, our experiments do not have this. A discussion and a few recommendations close the section.

### 4.1. Input Data

As input, we used *territorial data* and *GeoReports*. The first ones were necessary for the activation of the proposed *Service*. Specifically, they concerned:

- the Abruzzo administrative subdivisions. The source of the territorial data (in the shapefile format) used in the study is the *Archive of administrative unit boundaries for statistical purposes* provided by the *Italian Institute of Statistics*. The geometry of Abruzzo's municipalities and provinces was stored in the `subdivisions` table;
- data about assets and infrastructures inside the region. The assets that were examined in the study concerned: hospitals, schools, universities, public offices, dwellings, churches, banks, and malls; while the infrastructures that were taken into account concerned: roads, highways, and bridges. These data were extracted from OpenStreetMap Data (GIS Format Free shapefiles, SRS WGS84, Version 2017-03-02, available for free at http://download.geofabrik.de/). Assets and infrastructures were stored in the homonymous tables of the Geo-DB.

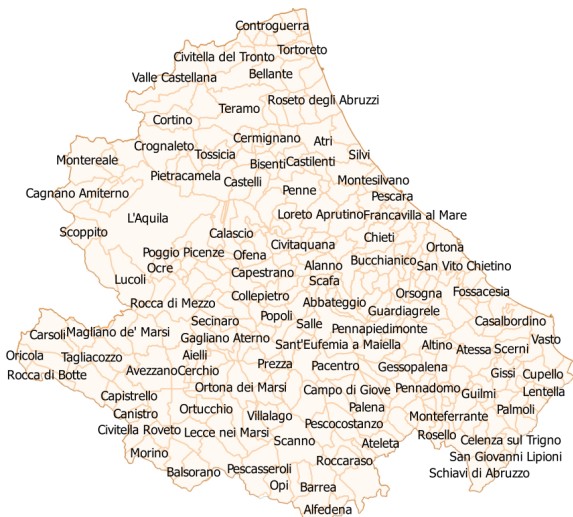

**Figure 14.** The municipalities of the Abruzzo region.

The thesaurus of synonyms about the assets and infrastructures located in the Abruzzo region was added to the above data.

The task of writing *GeoReports* (using TwittEarth) was assigned to 15 students of the course called *Advanced Databases* (offered by the University of L'Aquila in the second year of the Masters in Information Engineering). Six (of the 15) students were residents in the city hosting the University (L'Aquila), 3 in the homonymous municipality, 3 in the province of L'Aquila, and 3 in the remaining 3 provinces of the Abruzzo region (i.e., Teramo, Chieti, and Pescara). Each student was asked to write reports when he/she was in his/her city of residence and with reference to targets located there. This hypothesis reproduces the post-earthquake situation. A few weeks before starting the experiment (16 September 2018, the hypothetical time when the earthquake occurred), students were introduced to the scenario. They were told to imagine that they were part of a community hit by a severe earthquake. To help disaster responders to build a map about the municipalities hit by the event and the major damages suffered, they had to write *GeoReports* in the 24 h starting from 8:00 of 16 September 2018, about (hypothetically) damaged targets. Moreover, they were told to read carefully and comply with the guidelines listed below (which we called *the golden rules*):

- Refresh your knowledge about the name of sensitive assets and infrastructures in the town where you live. In case of a severe event, it is likely you will be there;
- Refresh your knowledge about the names of streets, roads, highways, and bridges that link the town where you live to the "world";
- Install TwittEarth on your smartphone;
- Become confident in the use of TwittEarth;
- Use the `#defaultHashtag` when sending a report about a damage;
- Send *GeoReports* about collapsed targets, only;
- When sending *GeoReports* about damaged targets, remember to activate the GPS option;
- Write and send your reports when you are as close as possible to the (hypothetically) damaged target;
- Refrain from sending multiple reports about the same target. Once is the best choice;
- Attach a photo of the target;
- Write the name of the target and its address (if any). Pay attention about these things, because only correct data will give rise to correct information.

The 15 students also contributed to the construction of the thesaurus, each one limited to the targets present in the administrative subdivision of residence. This strategy has two fundamental motivations: (a) distribute the workload among students; (b) benefit from their direct knowledge of the territory and the usual synonyms and abbreviations adopted by the citizens living there.

The 15 students wrote a total of 150 *GeoReports* over the 24 h of the simulation. As already pointed out, for the problem we studied, the reliability of the citizens' reports is the priority, while their number is not relevant, differently from what happens, for instance, in "sentiment analysis", where mining the text of one million tweets is much more meaningful than processing a few thousands of them.

Based on the reports sent by the students, the Service Manager (specifically, the authors of this paper) generated a certain number of *views* distributed along the hours immediately following the earthquake. Each view constitutes, through maps and tables, a snapshot of the situation in the territory.

### 4.2. The Views

**First assessment.** The first step of the simulation elaborated the 14 *GeoReports* sent within the first hour after the earthquake. The *Service* was responsible for identifying these reports from the Twitter repository and for copying them in the underlying Geo-BD. The execution (requested by the Service Manager) of pre-set SQL queries produced, as a final effect, the maps of Figures 15 and 16. They show, in order, the damaged municipalities (i.e., L'Aquila, Torniparte, Scoppito, Crognaleto, San Pio delle Camere, and Avezzano) and *the number* of warnings written by citizens (in order: 9, 1, 1, 1, 1, 1). The second map expresses (by means of colors) the ranking according to the number of *GeoReports* (the darker the color, the higher the municipality's score).

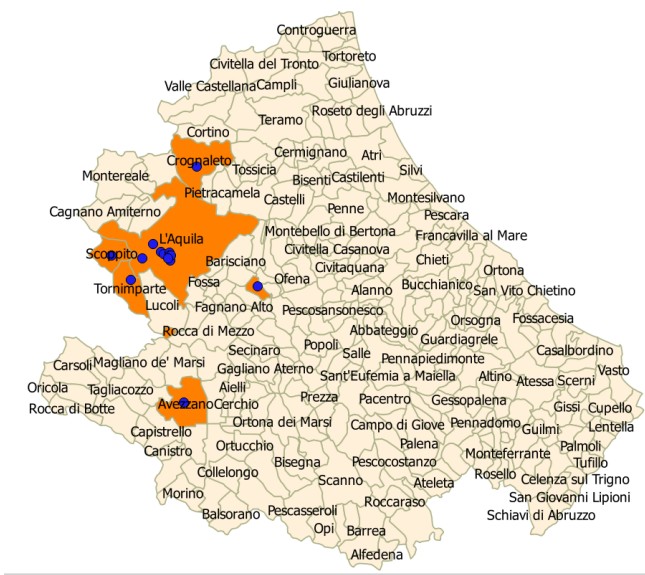

**Figure 15.** The census of damaged municipalities (within the first hour).

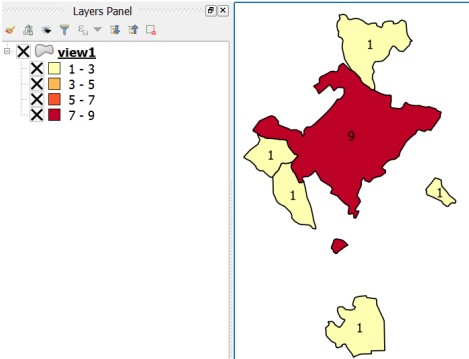

**Figure 16.** The ranking of the municipalities according to the number of retrieved *GeoReports* (within the first hour).

**Second assessment.** The second step of the simulation elaborated the 35 *GeoReports* sent between 9:00 and 10:00. The maps of Figures 17 and 18 show, in order, the damaged municipalities and the *total number* of *GeoReports* after two hours since the earthquake. The comparison between the two maps built at the end of the first and second assessment shows that the number of damaged municipalities increased from 6 to 19.

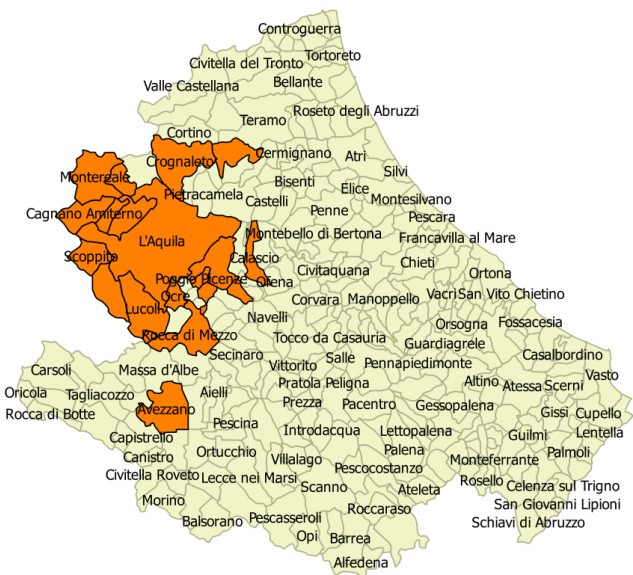

**Figure 17.** The census of the damaged municipalities (between the first two hours).

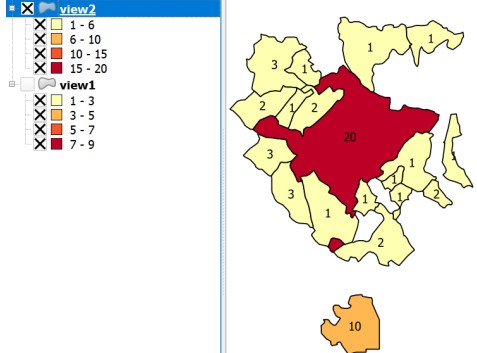

**Figure 18.** The ranking of the municipalities according to the number of retrieved *GeoReports* (in the first two hours).

Besides maps, it is convenient to propose to the disaster responders tables like those of Figures 19 and 20. The first of them shows the ranking of the top-11 municipalities affected by the earthquake with respect to the *score* parameter (i.e., to the number of reports that the *TwitterBridgeApplication* has extracted from the Twitter repository). Figure 20 proposes the ranking of damaged targets inside the municipality of L'Aquila. As is shown in Figures 19 and 20, the content of the two tables is the result of the processing of two stored SQL views (`view2` and `view3`).

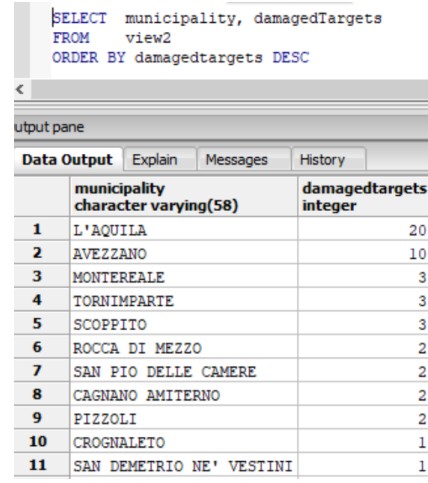

**Figure 19.** The ranking of the top 11 municipalities hit by the earthquake.

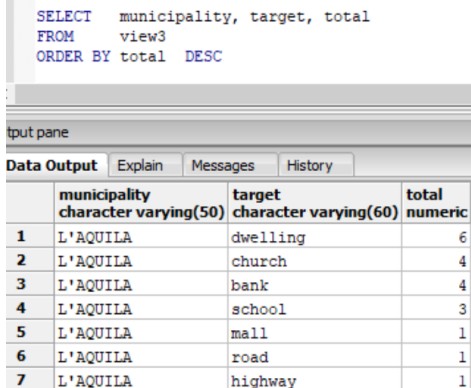

**Figure 20.** The ranking of damaged targets inside the municipality of L'Aquila.

Once the disaster responders have developed an overall picture about the territories hit by the earthquake and the most damaged categories of targets, they need to know the exact geographical location of those targets. This is the most important data for those who will have to intervene in the field to provide first aid and try to save lives.

By running queries that use the spatial SQL operators `ST_Buffer()` and `ST_Distance()`, it is possible to link the content of the citizens' reports with the descriptive and geometric content of the tables about the assets and infrastructures that are part of the Geo-DB. The result of these elaborations allows building maps that show the geographic position (i.e., the latitude-longitude pair) of damaged targets.

With reference to the municipality of L'Aquila and limited to the reports that were sent in the first two hours of the simulation, Figures 21–23 show, in sequence: (a) the map that brings together *all* the damaged targets; (b) the map about *all* the damaged dwellings, and finally, (c) the map about *all* the damaged bridges. Those maps are interactive, therefore selecting specific points on them, it is possible to read the geographical coordinates. Figure 24 shows the (partial) list of the damaged targets (as emerged by processing the reports written between 8:00 and 10:00) and their geographical position (as a pair of coordinates in the WGS84 SRS).

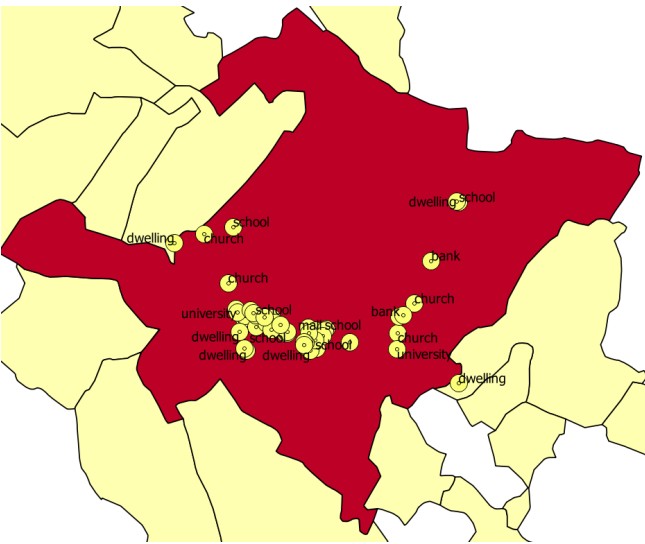

**Figure 21.** All targets in the municipality of L'Aquila damaged by the earthquake.

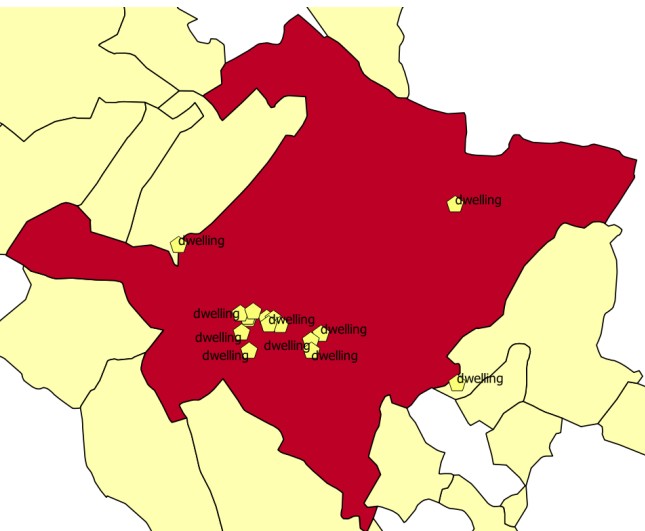

**Figure 22.** All dwellings in the municipality of L'Aquila damaged by the earthquake.

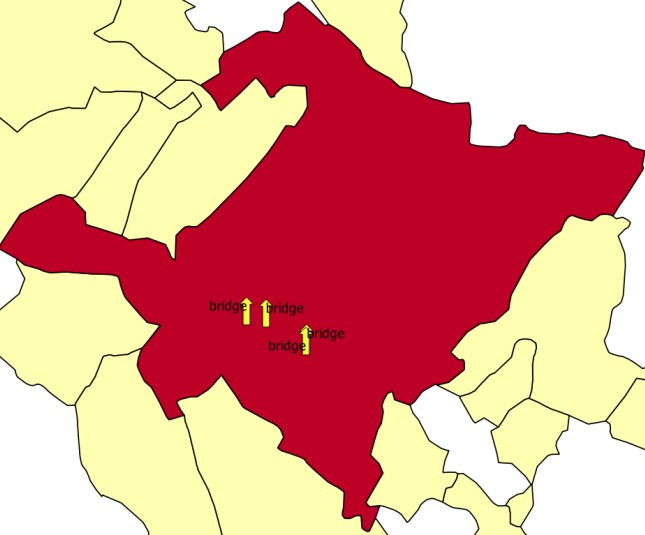

**Figure 23.** All targets in the municipality of L'Aquila damaged by the earthquake.

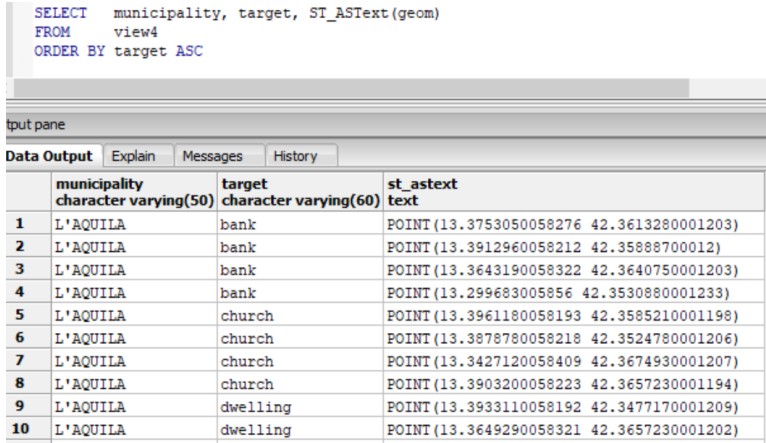

**Figure 24.** The damaged targets inside the municipality of L'Aquila and their geographical coordinates.

To increase the visual effectiveness of the maps that can be built using the proposed *Service*, it is sufficient to have recourse, as a background, to OpenStreetMap (http://www.geofabrik.de/geofabrik/), for example. Figure 25 presents three layers: OpenStreetMap, the buildings inside the city of L'Aquila, and the location of (some of the) damaged assets (the red circles).

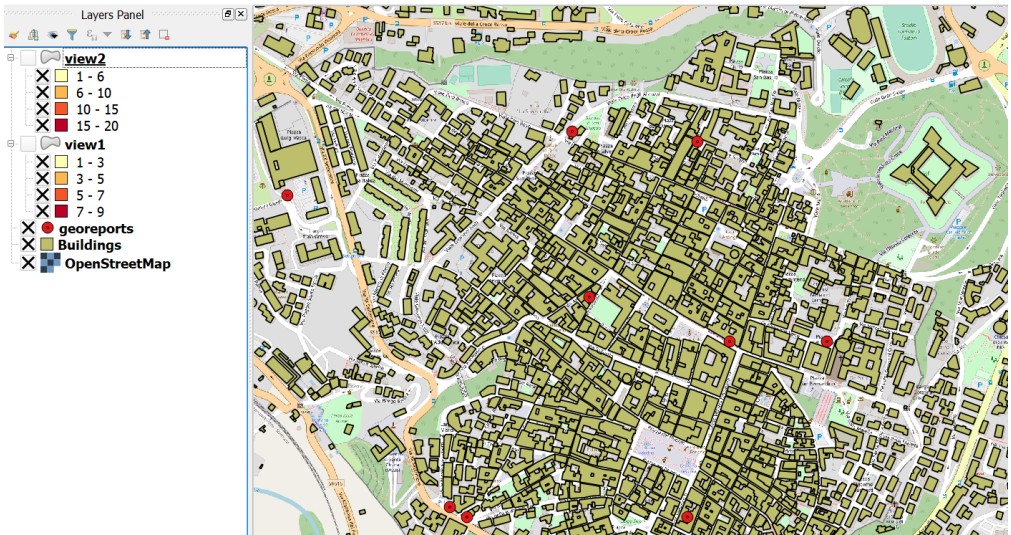

**Figure 25.** The location of few damaged buildings in the city of L'Aquila on top of an OpenStreetMap.

**Further assessments.** In the same way as described above, it is possible to generate other views of the situation in the territories hit by the earthquake.

The greater the temporal proximity between two consecutive assessments, the greater the adherence of the results returned by the TwittEarth app to the "on the ground" situation. About the number of assessments to be carried out, the issue is complex as they cannot be stopped as long as the *Service* receives warnings from municipalities not yet registered or concerning damages not yet reported. The control of the first condition is easy, unlike the latter, which imposes the adoption of a robust algorithm for the analysis of the text of the *GeoReports* and may not be sufficient, as noted in the section about the related work. As a practicable stop condition, it is advisable to stop assessments when the number of warnings coming from the subdivisions of territory $\mathcal{T}$ is drastically reduced.

*4.3. Discussion and Recommendations*

From the *friendly world* hypothesis, strictly respected by the 15 students involved in the case study, the following has been derived. The retrieval of the *GeoReports* from the Twitter repository was of 100%

for the presence of the keyword #defaultHashtag instantiated in the string #Laquila16Sept08:00 in all 150 reports sent by the students. Furthermore, the identification of the damaged target category between the two possible options (i.e., either *asset* or *infrastructure*) was 100%. In the same way, the data about the type of target damaged (for example *bank* instead of *school*) was unique and complete because the students had to choose a value in the predefined list proposed to them by the TwittEarth app.

In the actual use of the *Service*, the unambiguous identification of the specific target (i.e., either an asset or a hotspot along an infrastructure) and, consequently, the determination of its geographical position on the ground was a criticality that cannot be canceled nor estimated a priori (the underlying causes of this situation have been highlighted in Section 3.4). The high emotional stress that takes hold of people immediately after experiencing an earthquake that could have caused death and severe damages to the city where they live is certainly the primary factor that can make it difficult to remember the exact name or the complete address.

It is easy to foresee that the greatest number of reports whose processing may not be successful comes from citizens who live or work in large cities. In fact, in such cases, it is unlikely to assume that they memorized, and above all, to remember in the hours immediately after a severe earthquake the name and the address of the targets located in the territory. As illustrated in Section 3.4, the thesaurus plays a valuable role in the process of disambiguating names of targets and addresses. In this sense, it is essential for the thesaurus to be updated as much as possible and complete: two requirements that are difficult to satisfy in reality. The worst case takes place when the report sent by the citizen lacks the address of the damaged building. The developed *Service* reports the existence, in the Geo-DB, of unresolved *GeoReports*, leaving the Service Manager with the decision whether to view them personally or ignore them.

The automatic processing of the *GeoReports* is reliable if the messages contain *correct* and *complete* data about the targets damaged by the earthquake. In order to achieve this result, we believe that it is necessary to act in two independent phases. The first phase has to be anchored to structural initiatives such as training in (elementary and high) school and universities, about the active role that citizens of territories exposed to high seismic risk must assume when a severe event occurs, using social media in a responsible and correct way.

The other phase has to be activated in the imminence of an earthquake. The most frequent signal is a prolonged seismic swarm. For example, the earthquake of 6 April 2009 in L'Aquila (Central Italy) that killed 309 people (https://en.wikipedia.org/wiki/2009_L%27Aquila_earthquake) was preceded by a seismic swarm that, according to INGV (https://ingvterremoti.wordpress.com/2014/11/09/linizio-e-la-fine-della-sequenza-sismica-dellaquila/) began on 16 January 2009. This statement is based on the application of an algorithm proposed by Reasenberg [43].

About the short-term initiatives to promote the correct use of the TwittEarth app (or other similar) by citizens, Becker et al. [44] provided an interesting list of actions to be implemented both face-to-face and virtually. Virtual recruitment should include the activation of a dedicated website, as well as banner ads on local websites in addition to posting on Twitter and Facebook. It will also be effective to emphasize, in these periods in which the attention of the citizens is maximum, the *golden rules*, as well as the necessity of implementing them. This can be facilitated *if* the local authorities and/or the disaster responders in these warning periods conduct campaigns (in areas exposed to high seismic risk) to stimulate residents toward a broad and correct participation. The positive effects of the mechanism of *solicitation* were discussed in Section 2. The campaigns should, in particular, emphasize the importance that citizens know the names and addresses of the targets and places where they live and/or work; because it will be from there that they will have to send *GeoReports* in case a devastating earthquake manifests itself. To motivate citizens to do this "scholastic exercise", it will be appropriate that the campaign relies on the two further mechanisms mentioned in Section 2: *awareness of need* and *altruism*.

## 5. Conclusions

This paper is part of the rich group of studies that have investigated *how* to use social media to support the activities connected with the phases following a devastating event. Our contribution is based on the hypothesis of a *friendly world*, that is, a wide and correct participation by citizens on-site has been assumed. Under this assumption, a *Service* has been developed that involves two stakeholders: citizens and the Service Manager. Citizens have to send reports about assets and infrastructures that have suffered damages, using a mobile app (*TwitterEarth*) that guides them in writing the short report (at most 280 characters).

Today, apps are everywhere around us. The use of *TwitterEarth* is trivial. The positive side effect of using *TwitterEarth* is that the *Service* is able to capture *all* the pertinent messages rapidly. Restricting the attention to a limited number of messages that can be trusted is better than taking into account thousands of them with the tangible risk of extracting the wrong data.

The predefined structure of the *GeoReports* reduces the processing time of the *Service* (a critical factor in situations where the speed of decision-making and operations are of primary importance to save people), as both the extraction of the citizens' warnings from the Twitter repository and the analysis of their content are facilitated.

About the role of the Service Manager, it consists of invoking SQL views defined as objects on top of the geographical database, the heart of the entire *Service*. The GIS-oriented technological setting adopted to implement the *Service* is particularly suitable given the geographical dimension of the problem at hand.

Taking advantage of the *Service* proposed in this work, the Service Manager is able to make available to the disaster responders, in almost real time, maps and tables useful to get a picture of the areas most affected by the earthquake, the number of most damaged targets, and their geographical coordinates.

**Author Contributions:** Paolino Di Felice designed and formalized the method, performed the analysis, interpreted the data, wrote the manuscript, and acted as the corresponding author. Michele Iessi implemented the Service and the mobile app.

**Funding:** This research was funded by a grant from the University of L'Aquila.

**Acknowledgments:** The manuscript is the revised version of one previously submitted to IJGI. All the improvements we have done since that submission come from pertinent criticisms of two anonymous referees. We are infinitely grateful to them.

**Conflicts of Interest:** The authors declare no conflict of interest.

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
