# Peer review of "A Citizen-Sensing-Based Digital Service for the Analysis of On-Site Post-Earthquake Messages"

_ijgi, doi:10.3390/ijgi8030136_

Round 1

Reviewer 1 Report

The authors have made a great effort in revising the paper and now, in my opinion, it is ready to be published. Thanks for the patience and the work done.

Author Response

Thank you.

Reviewer 2 Report

The authors modified the paper according to the comments made to their previous submission of the paper to this journal. With these changes, this is a significantly improved version of the paper and I recommend it to be published in its present form.

Author Response

Thank you.

Reviewer 3 Report

Lines 65-75: Sec. -> Section. Section 1 presents the related work.

Line 147 - 149: statistics : total …, number …, percentage …, total ….

Line 271: vice versa -> on the contrary

Line 538: check population number

Line 564: Six (of the 15) … residents of …

Line 574: the first golden rule should be “do not expose yourself to danger” !!!

Line 599: Based on the reports

Figure 16 and Figure 18: According to cartographic theory, number of GeoReports should be portrayed with proportional symbols (point symbols) and not with areal symbols based on color. (https://mappingignorance.org/2013/12/16/the-complexity-of-drawing-good-proportional-symbol-maps/)

Figure 21: use a larger scale in this map

Line 642: figure 25 presents

Line 671: immediately after

Line 674: it is essential for the thesaurus to be

Line 679: In order that the results,

Line 682: elementary and high school

Line 685: the other phase ..

Line 689: if the local authorities

Author Response

Thanks for the suggested long list of improvements of the English.

All the suggestions have been implemented.

The suggestion of a possible variation of the layout of maps of Figures 16, 18 and 21 by adopting the so-called Proportional Symbol rendering type was ignored because the appearance of the maps is not the paper’s focus.

Geo-viewers on the market place allow the production of maps according to pre-defined personalized styles, so the System Manager may produce the same map in several different styles.

Reviewer 4 Report

The paper is highly improved with respect to the previous version (withdrawn by the authors). I recommend to carefully read the paper and to check the language with a native speaker.

I have only a minor comment: at line 146, please update the statistics according to the values available on the site the authors have cited.

Author Response

The statistics have been updated.

This manuscript is a resubmission of an earlier submission. The following is a list of the peer review reports and author responses from that submission.